# A Multi-Omics Characterization of the Natural Product Potential of Tropical Filamentous Marine Cyanobacteria

**DOI:** 10.3390/md19010020

**Published:** 2021-01-06

**Authors:** Tiago Leão, Mingxun Wang, Nathan Moss, Ricardo da Silva, Jon Sanders, Sergey Nurk, Alexey Gurevich, Gregory Humphrey, Raphael Reher, Qiyun Zhu, Pedro Belda-Ferre, Evgenia Glukhov, Syrena Whitner, Kelsey L. Alexander, Robert Rex, Pavel Pevzner, Pieter C. Dorrestein, Rob Knight, Nuno Bandeira, William H. Gerwick, Lena Gerwick

**Affiliations:** 1Center for Marine Biotechnology and Biomedicine, Scripps Institution of Oceanography, University of California San Diego, La Jolla, CA 92037, USA; tferreir@ucsd.edu (T.L.); nathan.a.moss@gmail.com (N.M.); raphael.reher@googlemail.com (R.R.); eglukhov@ucsd.edu (E.G.); swhitner@hawaii.edu (S.W.); k2alexan@ucsd.edu (K.L.A.); rrex@ucsd.edu (R.R.); pdorrestein@health.ucsd.edu (P.C.D.); wgerwick@health.ucsd.edu (W.H.G.); 2Skaggs School of Pharmacy and Pharmaceutical Sciences, University of California San Diego, La Jolla, CA 92093, USA; miw023@ucsd.edu (M.W.); rsilvabioinfo@gmail.com (R.d.S.); bandeira@ucsd.edu (N.B.); 3NPPNS, Physic and Chemistry Department, School of Pharmaceutical Sciences of Ribeirão Preto, University of São Paulo, Ribeirão Preto 14040-900, Brazil; 4Department of Pediatrics, University of California San Diego, La Jolla, CA 92093, USA; jonsan@gmail.com (J.S.); ghsmu414@gmail.com (G.H.); qiyun.zhu@asu.edu (Q.Z.); pbeldaferre@ucsd.edu (P.B.-F.); robknight@eng.ucsd.edu (R.K.); 5Center for Microbiome Innovation, University of California San Diego, La Jolla, CA 92093, USA; 6Cornell Institute for Host-Microbe Interaction and Disease, Cornell University, Ithaca, NY 14850, USA; 7Center for Algorithmic Biotechnology, St. Petersburg State University, St. Petersburg 199004, Russia; s.nurk@spbu.ru (S.N.); aleksey.gurevich@spbu.ru (A.G.); 8School of Life Sciences, Arizona State University, Tempe, AZ 85281, USA; 9School of Life Sciences, University of Hawaii at Mānoa, Honolulu, HI 96822, USA; 10Department of Chemistry and Biochemistry, University of California San Diego, La Jolla, CA 92093, USA; 11Department of Computer Science and Engineering, University of California San Diego, La Jolla, CA 92093, USA; ppevzner@ucsd.edu; 12Department of Bioengineering, University of California, San Diego La Jolla, CA 92093, USA

**Keywords:** genomics, metabolomics, marine cyanobacteria, natural products, biosynthetic potential

## Abstract

Microbial natural products are important for the understanding of microbial interactions, chemical defense and communication, and have also served as an inspirational source for numerous pharmaceutical drugs. Tropical marine cyanobacteria have been highlighted as a great source of new natural products, however, few reports have appeared wherein a multi-omics approach has been used to study their natural products potential (i.e., reports are often focused on an individual natural product and its biosynthesis). This study focuses on describing the natural product genetic potential as well as the expressed natural product molecules in benthic tropical cyanobacteria. We collected from several sites around the world and sequenced the genomes of 24 tropical filamentous marine cyanobacteria. The informatics program antiSMASH was used to annotate the major classes of gene clusters. BiG-SCAPE phylum-wide analysis revealed the most promising strains for natural product discovery among these cyanobacteria. LCMS/MS-based metabolomics highlighted the most abundant molecules and molecular classes among 10 of these marine cyanobacterial samples. We observed that despite many genes encoding for peptidic natural products, peptides were not as abundant as lipids and lipopeptides in the chemical extracts. Our results highlight a number of highly interesting biosynthetic gene clusters for genome mining among these cyanobacterial samples.

## 1. Introduction

Cyanobacteria are photosynthetic microbes that are abundant in a diverse range of habitats and support critical life processes in oligotrophic environments via photosynthesis and/or nitrogen fixation. Apart from their well-known importance in biogeochemical cycles because of their primary metabolic characteristics, cyanobacteria are also a prolific and sometimes distinctive source of secondary metabolites known as natural products (NPs). NPs have been a major inspirational source of new pharmaceutical agents [1]. Despite the advent of new bioprospecting techniques to mine NPs [2], the number of NPs discovered per year has remained relatively constant over the past decade [3]. The number of novel structures isolated has also remained constant [3], despite continuing investigation of under-explored habitats and microbial sources of NPs. As a complement to new microbial sources, multi-omics can be a powerful tool for prioritizing and directing the isolation of novel chemical entities from these organisms.

Notwithstanding the vast genetic diversity of cyanobacteria [4] and their different roles in human and planetary health, genomic investigations have been unevenly distributed throughout this phylum [5]. We observed that about 61% (1172 out of 1923) of the available National Center for Biotechnology Information (NCBI) strains that are from the unicellular genera *Prochlorococcus* and *Synechococcus* (subsection II), probably due to their ease of culturing, small genome sizes and their importance in oceanic nitrogen fixation and photosynthesis [6]. Moreover, organisms from these genera usually are less contaminated by associated heterotrophs in laboratory cultures. Finally, it has generally been found that members of these genera have a small number of biosynthetic gene clusters (BGCs, known for encoding the NP biosynthetic machinery in microbes). Altogether, these features tend to facilitate genome sequencing and assembly [7,8]. However, these genera are less relevant for genome mining and drug discovery efforts due to the low quantity and diversity of their BGCs. This contrasts with the genus *Moorea,* renamed to *Moorena* by Tronholm & Engene in 2019 [9], which on average, has a four-fold higher biosynthetic potential [10]. Reasons for the relative scarceness of sequenced NP rich cyanobacterial genomes may result from difficulty in culturing these types of filamentous marine cyanobacteria, the repetitive elements found in their genomes and their larger genome sizes.

In the present work, we took a multi-omics approach to profile the natural product potential harbored in 24 marine cyanobacterial genomes and the expressed molecules found in 10 of these (due to lack of material, not every sequenced sample could also be analyzed via liquid chromatography tandem mass spectrometry, LC-MS/MS,). Such a multi-omic approach is less common among cyanobacteria, with most studies being focused on individual molecules and their associated biosynthetic genes. Our genomic phylum-wide beta-diversity analysis highlighted the most promising BGCs and the most natural product diverse strains among cyanobacteria. Metabolomics revealed the major classes of molecules and library hits were dereplicated via spectral matching. Hence, we created a reproducible pipeline for omics profiling of microbes that performs a thorough dereplication and uncovers the most promising BGCs for subsequent genome mining.

## 2. Results

### 2.1. Diverse Sampling of Environmental Cyanobacteria

Overall, the SIO/CMBB tropical filamentous marine cyanobacteria collection consists of environmental samples collected and stored in RNA-later as well as 70 cultures grown in media and 95 stored on agar slants. For the current project, we sequenced 143 environmental samples, by Illumina HiSeq 4000, from genetically underexplored cyanobacteria, found as macroscopic tufts growing in sub-tidal tropical ecosystems around the globe (Appendix A, collection sites at Appendix A, sheet 1), along with 22 purified non-axenic cultures (total of 165 strains). Unfortunately, and most likely due to the issues described above, a majority of these sequenced genomes were either highly fragmented or very contaminated from heterotrophic bacteria, and therefore are not reported here. However, by applying the genome assembly steps outlined below to this combination of environmental and cultured cyanobacterial samples, we obtained 24 high quality draft genomes (over 90% CheckM completeness and less than 500 contigs). Of these, 19 derived from environmental samples and 5 from non-axenic mono-cyanobacterial cultures.

### 2.2. Genomics Assembly Pipeline and Phylogenomics

The current taxonomic distribution of 1923 publically available cyanobacterial genomes from NCBI RefSeq database (including our 24 samples) (May 2020) is heavily biased toward subsection II. Strains from this subsection tend to be less promising for drug discovery due to the low abundance and diversity of BGCs in their genomes [10]. Here, we developed a reproducible genomic pipeline for assembly and initial mining of cyanobacterial genomes. This pipeline sequentially performs assembly by metaSPAdes [11], taxonomic binning using GC content and DarkHorse [12] taxonomic assignments, and quality control analysis to yield 24 high quality genomes that can be used for more in depth investigations, such as biodiversity assessment and comparative genomics of cyanobacterial natural product BGCs. For one of our cultures, *Leptolyngbya* sp. SIOISBB (collection code ISB3NOV948B CUL), which is currently under development as a heterologous host for expression of marine BGCs, we complemented the short read sequencing with long reads from Nanopore MinION^®^ in order to obtain the complete genome. For a second culture, *Moorena* sp. SIO1ASIH (collection code ASI16JUL142 CUL), which is the producer of a related suite of compounds produced by a unique and combinatorial biosynthetic logic (vatiamides A-F) [13], we complemented the short read sequencing with PacBio RS^®^ in order to reduce the draft genome to 24 contigs. Lastly, for a third culture, *Leptolyngbya* sp. SIO1E4_02 (collection code ASX22JUL142CUL), we obtained 6 contigs using a hybrid assembly between HiSeq 4000 short reads and nanopore MinION^®^ long reads.

Cyanobacterial phylogeny has been and continues to be challenging, in large part due to previous assignments that used morphological characteristics and lacked a genetic basis. Figure 1 illustrates the phylogenetic assignments for the 24 high quality draft genomes reported herein, and includes 6 *Moorena*, 10 *Okeania*, 4 *Symploca*, 2 *Leptolyngbya*, 1 *Oscillatoria* and 1 *Spirulina*. As can be observed in Figure 1, the *Moorena*, *Leptolyngbya* and *Symploca* clades are tightly defined. In contrast, the *Okeania* clade appears to be more diverse (higher degree and number of branches) and its clade includes as the closest relatives *Trichodesmium erythraeum* IMS101 and *Trichodesmium* sp. LADK01. It appears that the genera *Okeania* and *Trichodesmium* arose from a recent common ancestor; this might explain why the latter genus appears inside the *Okeania* clade (Figure 1). MUMmer [14] genome alignments (Appendix A) between the two aforementioned *Trichodesmium* and two of the closest *Okeania* (SIO3I5 and SIO2F4) show that there is no clear distinction in the genetic architecture of *Okeania* and *Trichodesmium*. All plots in Appendix A exhibit good synteny, although, these four genomes contain many contigs, making it difficult to properly assess synteny. Additionally, these two genera are distinct regarding their previously characterized natural products (example, *Okeania* produces malyngamides and *Trichodesmium* does not) [15]. The addition of more *Trichodesmium* genomes to the NCBI database will help to construct a more complete picture of the differences and similarities between these genera and their phylogeny.

The 24 high quality draft genomes obtained in this study ranged in size from 7.1 to 9.8 Mbp, with the average size of 8.42 Mbp. The GC content varied from 36.8% to 52.1%. The number of scaffolds ranged from 3 up to 471. The total number of BGCs ranged from a genome with 10 BGCs up to an exceptional genome with 46 BGCs, an average of 21.1 BGCs per genome (see statistics in Table 1; for more detailed statistics and metadata on the scaffolded genomes, refer to Appendix A, sheet 1). We also performed a Mash [17] whole-genome comparative analysis between the 24 assembled strains and the 1899 cyanobacteria genomes present in the NCBI database, where the cutoff threshold was the same as reported by Ondov et al., 2016 [17] (distance ≤ 0.05 and *p*-value ≤ 10^−10^). We observed that 7 out of the 24 genomes (NCBI RefSeq ID includes JAAHFU01, JAAHHN01, JAAHGO01, JAAHGS01, JAAHFN01, JAAHGL01 and JAAHII01, genomes colored red in Table 1) were considered singletons in the analysis (i.e., did not network with any other genome from NCBI). The remaining 17 genomes matched to one of our lower quality draft genomes published in the NCBI database (we uploaded to NCBI the 51 additional draft genomes from other tropical marine cyanobacteria that are not part of this manuscript). For these 7 singleton genomes, this analysis indicates that they are “rare” compared to available entries in public databases, and hence they provide valuable insights on the metabolic capacities of a broader range of cyanobacteria as well as represent targets for future genome mining.

### 2.3. Genome Mining Pipeline

The 1923 cyanobacterial genomes analyzed in this phylum-wide study were scanned for BGC potential using antiSMASH v5.1.2 [18]; this identified a total of 12,323 BGCs. These BGCs were compared via domain similarity using BiG-SCAPE v1.0 [19] which groups homologous BGCs into gene cluster families (GCFs). The networking threshold was selected by performing a cutoff calibration using BGCs from the MIBiG database [20]. For such, we used families containing two or more previously characterized BGCs from the MIBiG database (containing 2057 BGC entries) [20]. Using Tanimoto scoring [3], we performed a comparison of the annotated structures and structural similarity between the MIBiG gene pathways. By inspecting different cutoffs, depicted on the x axis of Figure 2a, we observed the average Tanimoto structural similarities (Figure 2a, y axis) between two or more MIBiG BGCs that were networked together in the same family. Hence, we observed that the best BiG-SCAPE similarity score cutoff was 0.7 (0.3 distance in Figure 2a). This cutoff selected during the MIBiG validation was then applied to the remainder of the dataset, generating a gene cluster similarity network. In this case, the gene cluster network contained 7085 BGCs from the entire cyanobacteria phylum (several BGCs did not network under the selected networking conditions) plus 2057 BGCs from MIBiG, yielding 1084 gene cluster families (GCFs).

Because 274 genomes only had singletons and thus they did not network using the selected cutoff, a total of 1649 genomes could be networked. We then considered which classes of GCFs predominate among all cyanobacteria, as well as which classes of GCFs are found in our 24 marine cyanobacteria. This analysis revealed that the category of “Unknown” biosynthetic class predominated among the full cyanobacteria genome dataset (Figure 3a), followed by bacteriocins, nonribosomal peptides, and terpenes. Deeper manual interrogation of these unknown GCFs revealed that while their BGCs are very different from known BGCs, they do use the same broad enzymatic families and thus are recognizable as encoding for NPs. This underscores the observation that the broad set of cyanobacteria have many BGCs that are still poorly understood, and will likely yield the discovery of many novel natural products. Interestingly, the same analysis of the 24 marine cyanobacteria being newly reported herein do not have the category “unknown” as their major GCF type. Rather, NRPS metabolites, bacteriocins/cyanobactins, terpenes and type I PKS products, as well as hybrids, are the major classes in these filamentous marine cyanobacteria. As it is uncertain what the “unknown’s” truly represent in these analyses, it remains unclear if fundamentally new types of scaffolds still await discovery. From another perspective, the GCF categories present from the phylum wide analysis generally match those in the 24 marine samples described herein, except for the categories hybrid bacteriocin-lanthipeptide, cyclodipeptide synthase (CDPS) and ectoine.

Next, using the same gene cluster similarity network, we accessed the distribution of GCFs in these samples and calculated diversity scores, including beta-diversity using Jaccard similarity. In this analysis, beta-diversity illustrates the percent difference between pairs of samples in terms of their shared BGCs (pairwise measurement defined as the intersection over the total of the two cyanobacterial strains A and B). In a phylum-wide analysis (1923 cyanobacterial genomes and 1650 networked genomes), cyanobacteria that exhibited a low average beta-diversity and a small number of BGCs in their genomes appeared to be the most likely to yield the rediscovery of previously characterized NP scaffolds. A Principal Coordinate beta-diversity analysis (PCoA plot from Figure 2b) clearly identified a group that harbors the most diverse distribution of GCFs, containing an average dissimilarity score of over 95%. All 24 marine cyanobacteria here investigated belonged to the group with high beta-diversity, confirming that tropical filamentous marine cyanobacteria are indeed a good source for the discovery of new metabolites. Moreover, this analysis reveals that there are an additional 630 genomes (Figure 2b, samples colored in blue and genomes listed in Appendix A, sheet 3) from under-explored cyanobacteria that could be rich sources of novel natural product scaffolds.

Additionally, we created a rarefaction curve for the presence/absence of GCFs in these genome-sequenced cyanobacteria. Appendix A illustrates that the slope of the fitted curve decreases considerably with an increasing number of sequenced genomes, indicating that not many more cyanobacterial genomes are necessary in order to sample the total diversity of cyanobacterial BGCs (e.g., to reach a slope close to zero). Nonetheless, 61% (1172 out of 1923) of the available NCBI strains are from the unicellular genera *Prochlorococcus* and *Synechococcus* (cyanobacteria with few BGCs per genome) and that might bias this analysis. This bias is also suggested by the fact that the beta-diversity analysis identified many under-explored cyanobacteria. However, removal of *Prochlorococcus* and *Synechococcus* genomes from this analysis (Appendix A) resulted in the same general trend (this second curve also reaches a plateau). More sequencing of under-explored cyanobacteria is necessary to clarify if the rarefaction curve for GCFs is truly reaching a plateau.

### 2.4. Metabolomics Pipeline

We collected high-resolution LC-MS/MS spectra so as to examine the metabolomes of a subset (10 of 24) of these genome sequenced tropical marine cyanobacteria. We analyzed the MS/MS spectra obtained in our runs using the well-established automatic annotation metabolomics platform Global Natural Products Social Molecular Networking (GNPS) [21]. The GNPS platform (http://gnps.ucsd.edu) harbors several tools, including the following used in the current study: (1) classical molecular networking with dereplication via spectral match, generating predicted structures; (2) MolNetEnhancer [22] that includes the *in silico* tools Network Annotation Propagation [23] and DEREPLICATOR+ [24], both also predicting structures for the molecules without spectral match; (3) followed by chemical class annotation of the predicted structures with ClassyFire [25] (incorporated to the MolNetENhancer pipeline). (4) Download of the highly annotated network and visualization in Cytoscape [26]. As observed in Figure 4, the number of annotated nodes in the MolNetEnhancer network was not very high but the use of *in silico* tools significantly improved the number of predictions made via GNPS. The annotation rate via spectral match was only 4.7% (68 out of 1442 nodes were annotated, a third of the annotation rate found in public GNPS datasets). However, *in silico* tools annotated 87 out of 96 molecular families (only 9 families are colored red for “no match”, singletons not counted). This indicates that these expressed metabolites belong to known structure classes that have already been well explored in these cyanobacterial genera. For every spectral match, we manually evaluated each MS/MS mirror plot to validate the library hit. We excluded from the network (Figure 4) common contaminants and the annotated molecules that were also present in the LC-MS/MS blanks. We observed several genus specific molecular families (names in red in Figure 4). Predicted molecules without a common name (for example, common names like veraguamide, apramide, dolastatin and so on) were labeled by their most detailed ClassyFire prediction. Full International Union of Pure and Applied Chemistry (IUPAC) names can be found at Appendix A, sheet 2 and predicted structures in Appendix A). These genus specific molecules included veraguamide A and K [27], “clerodane diterpenoid”, “chromone” and “8-O-methylated flavonoid” in *Okeania*; tumonoic acid G and I [28], lyngbyastatin 3 [29], dolastatin D [30] and hoiamide A-B [31,32] in *Symploca*; apramide A [33], carriebowmide [34], “ingenane diterpenoid” and “dihydroxy bile acid” in *Moorena* (see all library hits in Appendix A, sheet 2 and for predicted structures for the molecular classes in quotations, see Appendix A). Several other molecules isolated from *Moorena* are present in the GNPS database, however, it appears that they were not detected in our samples, except for barbamide [35] (a genus specific compound that was a singleton classified as “no match” and therefore it was not included in Figure 4). *Leptolyngbya* had no genus specific metabolites annotated in the GNPS database. While there are several metabolites known to be produced by *Leptolyngbya* [36], only six of them are present in the GNPS database (dolastatin 12 [37], palmyrolide A [38], phormidolide [39], scytonemin [40], shinorine [41] and stilbene [42]), and these were not detected in our analyses. *Spirulina* and *Oscillatoria* were not sampled for LC-MS/MS.

## 3. Discussion

The Mash whole-genome comparison indicates that 29% of the genomes that we sampled in this study are unique (compared to genomes in NCBI) in terms of their genomic content, including BGCs. This result is consistent with our previous reports that the genus *Moorea* (renamed to *Moorena*) is distinctive from other cyanobacteria, especially regarding its natural product potential [7]. Other genera included in this study, such as *Okeania*, *Leptolyngbya* and *Oscillatoria*, have also been reported as abundant producers of natural products [36,43]. This contrasts with the genus *Spirulina* which is not reported as a prolific metabolite producer. The hypothesis that the former cyanobacterial genera represent an untapped natural product potential was confirmed by our presence/absence BGC beta diversity analysis. This analysis included all 1649 cyanobacterial genomes, and indicated that all 24 of the marine cyanobacterial genomes presented here possessed high average beta diversity. The major classes observed among the GCFs found in the 24 marine cyanobacteria were peptides (NRPS, bacteriocins and cyanobactins), terpenes, lipids (T1PKS) and lipopeptides (NRPS-T1PKS hybrids). These results taken together indicate that these 24 marine strains have a vast and under-explored genetic potential for the discovery of new natural products, and the high abundance of NRPS BGCs are a good target for genome mining.

MolNetENhancer (including molecular networking) was useful for exploring three significant issues. First, what previously known or highly related molecules are present in these samples? Secondly, are there genus specific molecules? And finally, are the classes observed in our samples consistent with the literature and consistent with known GCF classes? Interestingly, we did not observe many peptidic NPs in the metabolomic analysis, in contrast to their abundance in the BGC analysis (RiPPs and NRPS). Rather, by metabolomic analysis, we observed many more lipid-like molecules (Figure 4). This might have one of a number of reasons. For example, it is possible that the peptide NPs are excreted (diketopiperazines have been reported as signaling molecules) [44], expressed in low quantities below our detection limits, or are not as efficiently extracted given our protocols which favors lipid-like molecules. Alternatively, it may be that these peptides are simply not expressed under the conditions of growth at the time of extraction whereas the NRPS-T1PKS and T1PKS metabolites are being expressed.

The fact that a majority of the molecular families revealed by the mass spectrometry analysis (90%) had a spectral match or an *in silico* annotation suggests that discovery of additional novel scaffolds in these 10 samples will be limited. However, this genetic analysis also indicated that many of these same GCFs are under-explored in terms of the actual NP that is produced. One could envision that as heterologous expression of cyanobacterial natural products becomes more reliable and accessible (i.e., we recently expressed a large marine cyanobacterial pathway in *Anabaena* PCC 7120) [45], these rare BGCs from diverse marine strains can be targeted for heterologous expression, isolation and structural characterization of novel natural products, and pharmacological description.

In conclusion, we were able to enrich the diversity of genomic information for natural product rich cyanobacteria by providing 24 new high quality draft genomes, 7 of which appear to be highly unique compared to those in the current NCBI database. We demonstrated via a phylum-wide analysis that prioritization of samples using beta-diversity can highlight “natural product diversity hotspots” in a given dataset. Our metabolomic analysis revealed several of the most abundant metabolic classes that are expressed and retained in the biomass of these samples. The DNA sequence and metabolomics information generated in this study improves understanding of the natural product potential of tropical marine cyanobacteria and enables prioritization of samples for a genome mining approach to natural product discovery.

## 4. Materials and Methods

### 4.1. Collection, DNA Extraction and Sequencing

Samples were collected via SCUBA diving or snorkeling in shallow benthic environments (less than 20 meters) from different coastlines around the globe (Appendix A). In each case, the collected biomass was preserved in RNA-later for subsequent sequencing or 1:1 isopropanol:seawater for mass spectrometry analysis. Given that *Moorena* and *Okeania* can form macroscopic tufts in seawater, we focused on collecting samples that matched the morphology from these two genera. When possible, we obtained purified cultures using standard microbiology and microscopy techniques, generating 4 non-axenic mono-cyanobacterial cultures used in this study [46]. RNA-later samples and cultured biomass were processed via freezing with liquid nitrogen, followed by grinding and extraction according to the “QIAGEN Bacterial Genomic DNA Extraction Kit” protocol, incubation times extended for 1 h and the volume of Proteinase K used was 10 µL (10 mg/mL). For details on the extraction procedure, see reference [46].

DNA libraries were generated using a miniaturized version of the Kapa HyperPlus Illumina-compatible library prep kit (Kapa Biosystems^®^, Wilmington, MA, USA). DNA extracts were normalized to 5 ng total input per sample in an Echo 550 acoustic liquid handling robot (Labcyte Inc, San Jose, CA, USA). Next, we used a Mosquito HTS liquid-handling robot (TTP Labtech Inc, Melbourn, United Kingdom) for 1/10 scale enzymatic fragmentation, end-repair, and adapter-ligation reactions. Sequencing adapters were based on the iTru protocol [47] in which short universal adapter stubs are ligated first and then sample-specific barcoded sequences added in a subsequent PCR step. Amplified and barcoded libraries were then quantified by the PicoGreen assay and pooled in approximately equimolar ratios before being sequenced on an Illumina HiSeq 4000 instrument to >30X metagenomic coverage. Two samples were complemented with long read sequence. *Leptolyngbya* sp. SIOISBB was sequenced with Nanopore MinION^®^ using 1D² Sequencing Kit (R9.5) (ligation-based). *Moorena* sp. SIOASIH was sequenced PacBio RS^®^ using a 10 kb library prep. *Leptolyngbya* sp. SIOASX libraries were prepared using the Genomic DNA by Ligation kit (SQK-LSK109, Oxford, UK), incubation times during ligation attachment were extended to 20 min at 37 degrees Celsius, and “Long Fragment Buffer” was used during final AMPure bead purification to preserve DNA fragments longer than 3 kb. Libraries were then sequenced for 48 h using the Nanopore MinION^®^ (Flow Cell R9.4.1) with flow cells being flushed and reloaded with additional material from the library to be sequenced for more 24 h.

### 4.2. Genome Assembly Pipeline

The 24 metagenomic samples were assembled with metaSPAdes 3.12.0 [11]. Assembled contigs were annotated with Prokka 1.11 [48] and the phylogenetic assignment for each annotated gene was predicted via DarkHorse 2.0 [12,49]. Only contigs that followed minimal requirements were binned in: one or more cyanobacterial genes, and; GC content smaller or equal to 58% (determined via QUAST analysis [50] of high GC draft metagenomes). The 24 draft genomes were successfully scaffolded using MEDUSA [51] and high quality reference genomes from the NCBI database. Once scaffolded, the quality control via CheckM [52] approved all 24 high-quality draft genomes (over 90% completeness). *Leptolyngbya* sp. SIOISBB long reads were process for base calling with Albacore 2.0.2 (https://github.com/dvera/albacore), trimmed with PoreChop 1.0 (https://github.com/rrwick/Porechop) and assembled with Canu 1.6 [53]. *Moorena* sp. SIOASIH long reads were part of a hybrid assembly with metaSPAdes 3.12.0, followed by binning and quality control as described above for the short reads. *Leptolyngbya* sp. SIOASX reads were processed using Unicycler Hybrid Assembler [54] and the assembled metagenome was then binned using MetaBat2 (https://bitbucket.org/berkeleylab/metabat/src/master/) and each bin was given a phylogenetic assignment and checked for completeness using CheckM [52]. Mash [17] was used to compare whole genomes.

### 4.3. Phylogenomics

We selected the same set of conserved 29 housekeeping genes used in Leao et al. (2017) [7] and Calteau et al. (2014) [16] for phylogenomic analysis. The homologs of these housekeeping genes were identified in our strains via Diamond search v0.8.31.93 [55]. Once identified and extracted, the set of 29 genes were aligned using MUSCLE v3.8 [56] and trimmed via trimAl v1.2 [57] (both in standard settings). Once the alignments were complete, we concatenated the housekeeping genes. A phylogenetic tree was reconstructed based on the concatenated ribosomal protein sequences extracted from the cyanobacterial genomes, using maximum likelihood (ML) implemented in IQ-TREE 1.6.10 [58]. Amino acid substitution model was determined using ModelFinder [59] as part of IQ-TREE, which chose LG + R10 (LG substitution matrix, plus FreeRate model with 10 rate categories) as the best model. Phylogenetic reconstruction was performed using this model and IQ-TREE default settings. Branch supports were provided using 100 replicates of classical bootstrap, and the out-group was *Melainabacteria* SM1 D11. The overall shape and clades of the tree were consistent with previous studies. The genomes selected to be part of the tree were as follow: the set of 107 complete genomes from Leao et al. (2017) [7], 51 low quality marine filamentous cyanobacterial draft genomes published at NCBI, and the 24 high quality tropical marine cyanobacterial draft genomes reported in this study.

### 4.4. Gene Cluster Networking and Diversity Analysis

The 24 high quality scaffolded drafts (along with 1649 cyanobacterial genomes that we were able to download from NCBI) were investigated for their biosynthetic gene clusters (BGCs). Their predicted BGCs via antiSMASH v3.0.5 [60] were networked using BiG-SCAPE [19] to obtain pairwise similarity scores between the BGCs. The similarity cutoff calibration was determined by checking the networking performance for annotated BGCs from the MIBiG database (Minimum Information about a Biosynthetic Gene cluster database [20]). As observed in Figure 2a, the best cutoffs included 0.7 similarity (0.3 distance), which corresponded to 99% structural similarity between the annotated metabolites in MIBiG. By selecting a cutoff, we converted a similarity metric into presence/absence of gene families (group of homologous BGCs). We built a pairwise Brays-Curtis beta-diversity dissimilarity matrix among all tested strains (using skbio.diversity package in python, http://scikit-bio.org/), including the 1649 previously published NCBI cyanobacterial genomes. Subsequently, we calculated the average Brays-Curtis beta-diversity per strain and built a Principal Coordinate Analysis (PCoA) plot to highlight samples with high diversity scores (over 95% average Brays-Curtis beta-diversity). A custom python script identifying gene cluster families (GCFs) as present/absent was created and all different GCFs were summed to produce the rarefaction curve in Appendix A.

### 4.5. Extraction and UHPLC-MS/MS Analysis

We generally utilized a similar analysis by UHPLC-MS/MS as described in Luzzato-Knaan et al., 2017 [61]. Ten (10) biomass samples preserved for chemical extraction were extracted 5 to 8 times with CH_2_Cl_2_:MeOH 2:1, and dried in vacuo. The chemical extracts were analyzed with an UltiMate 3000 UHPLC system (Thermo Scientific, Waltham, MA, USA) using a Kinetex 1.7 mm C18 reversed phase UHPLC column (50 × 2.1 mm) and Maxis Impact Q-TOF mass spectrometer (Bruker Daltonics, Billerica, MA, USA) equipped with an Electrospray Ionization (ESI) source, run in the positive mode. Injections were made of 3.3 µg per sample. The chromatographic gradient was 5% solvent B (ACN/H_2_O/formic acid 98%/2%/0.1%) with solvent A (H_2_O/ACN/formic acid 98%/2%/0.1%) for 1.5 min, a step gradient of 5% to 50% B in 0.5 min, held at 50% B for 2 min, a second gradient of 50–100% B in 6 min, held at 100% B for 0.5 min, 100–5% B in 0.5 min and kept at 5% B for 0.5 min at a flow rate of 0.5 mL/min throughout the run. MS spectra were acquired as previously described in Garg et al., 2015 [62].

### 4.6. GNPS Molecular Networking

We used classical molecular networking [21] combined with the *in silico* tools Network Annotation Propagation [23] and DEREPLICATOR+ [24] to annotate structures that were the query for the chemical ontology tool ClassyFire [25]. Parameters for the molecular networking were a cosine equal to or greater than 0.7, minimum matched fragment ions of 4, minimum cluster size of 2, library search minimum matched peaks of 4 and search analogs turned off.

The classical molecular network job is available at https://gnps.ucsd.edu/ProteoSAFe/status.jsp?task=9c4f6a15087743f7a97907c3229c4711 and MolNetEnhancer network job is available at https://gnps.ucsd.edu/ProteoSAFe/status.jsp?task=80716381d0694b80aa395c750f1f21dc; the job can cloned for reproduction purposes. The final network is also available at the folder outputs in the GitHub repository.

The structures from Appendix A come from enhancing a GNPS classical molecular network with MolNetEnhancer. Classical molecular networking compare molecules that have a fragmentation spectrum with themselves and with the molecules in the GNPS library (generating library spectral matches). For enhancing a classical molecular network, we separately ran (for the same set of LC-MS/MS files) the tools for *in silico* prediction, Network Annotation Propagation and Dereplicator+. Once these two *in silico* analysis were completed, we ran MolNetEnhancer that uses ClassyFire on the predicted structures from both GNPS library hits and the *in silico* tools to derive the chemical ontology for each structure. ClassyFire uses structural features to automatically annotate a molecular structure into one of the over 4800 different categories, including predictions that vary from Kingdom, SuperClass and so on until fairly detailed categories. MolNetEnhancer produce an output-file that contains a network very similar to Figure 4, with node colors pre-populated according to the predicted ClassyFire SuperClass that was inferred using the aforementioned structures. The nodes table of the MolNetEnhancer network contains the full ClassyFire predicted ontology, predicted structure and other GNPS information. This node table was pasted in Appendix A, sheet two.

Our work can be reproduced by accessing the jupyter notebooks (in the GitHub repository), cyanobacterial genomes (all deposited at the NCBI database), the LCMS files (in the MassIVE database) and the links for the network jobs (in the GNPS platform).

## Figures and Tables

**Figure 1 marinedrugs-19-00020-f001:**
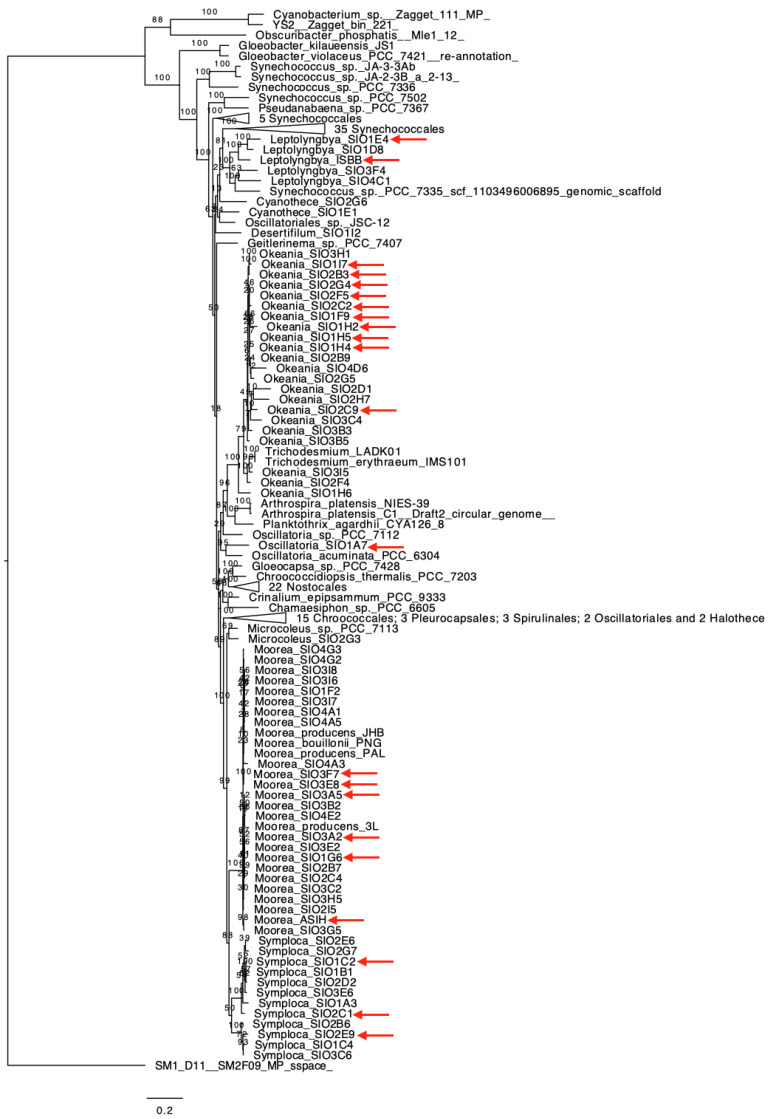
Phylogenomic analyses of completed cyanobacterial genomes using 29 conserved genes from Calteau et al. [16]. Tips were labeled either according to phylogenomic cladding and 16S rRNA identity, where our 24 high quality genome are indicated by red arrows. Bootstrap values are labeling the branches. *Moorea* was renamed to *Moorena* in 2019; however, because the NCBI records are still labeled as *Moorea*, we use the latter name in this phylogenomic tree.

**Figure 2 marinedrugs-19-00020-f002:**
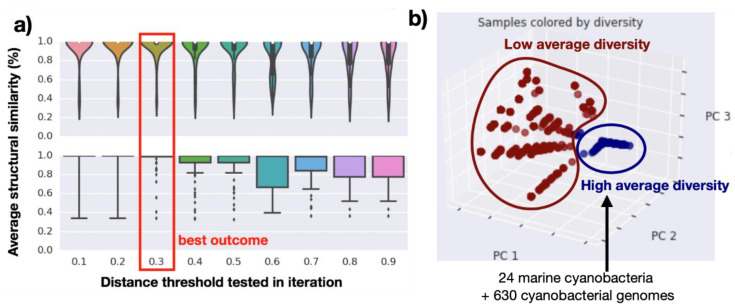
(**a**) Cutoff comparison networking the expert-annotated biosynthetic gene clusters from MIBIG database, highlighting the best outcome for the cutoff of 0.3 distance (70% similarity). Top violin plot illustrates the distribution of the average similarity scores and bottom box plot also illustrates these scores but focusing on the number of outliers per cutoff. (**b**) Principal Coordinate Analysis (PCoA) for beta-diversity scores from 1650 genomes that were networked (including the 24 marine cyanobacterial genomes generated in this project). The blue group represents the most diverse samples in the dataset (samples for which the beta-diversity was over 95%; includes a total of 654 cyanobacterial genomes including the 24 reported herein; genomes listed at Appendix A, sheet 3) and the red group represents low diversity samples (genomes with an average beta-diversity score below 95%; includes a total of 996 cyanobacterial genomes).

**Figure 3 marinedrugs-19-00020-f003:**
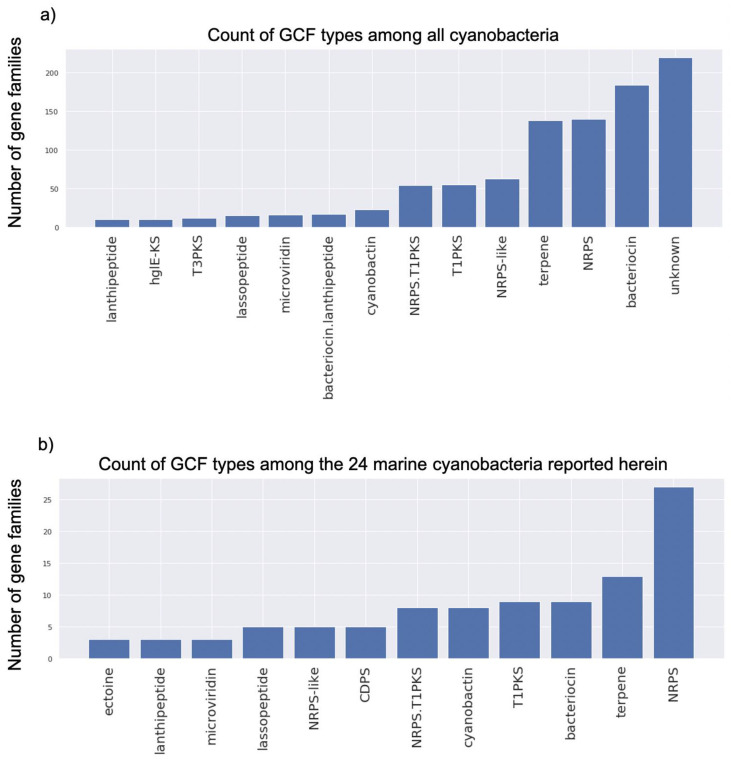
(**a**) Histogram counting all gene cluster families separated by common biosynthetic class (classification according to BiG-SCAPE and cutoff of at least 10 occurrences in all cyanobacteria). (**b**) A similar histogram to panel A using only the 24 marine cyanobacteria newly reported herein and with a cutoff of at least 2 occurrences in these cyanobacteria. NRPS = nonribosomal peptide synthetase; T1PKS = type 1 polyketide synthase; T3PKS = type 3 polyketide synthase; CDPS = cyclodipeptide synthase; hglE-KS = heterocyst glycolipid synthase.

**Figure 4 marinedrugs-19-00020-f004:**
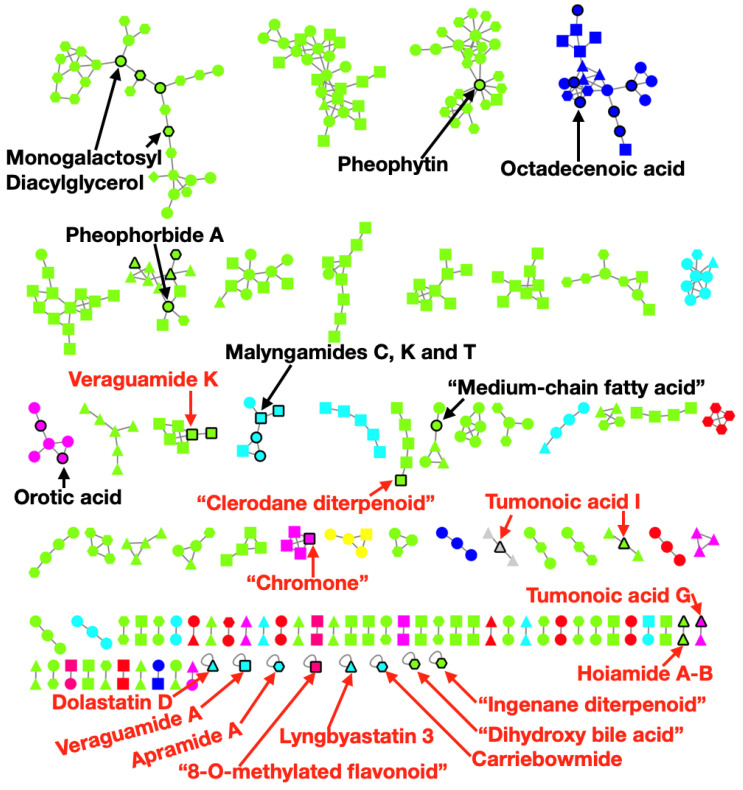
Classical molecular networking improved via *in silico* tools and MolNetEnhancer. Self-loops (singletons) annotated as “no match” were removed. Nodes are colored according to the ClassyFire predicted Superclass. Nodes with black border represent library hits and one representative per molecular family is displayed in the network. Genus-specific families with library hits are labeled in red. Diamonds represent *Leptolyngbya* samples; hexagons represent *Moorena*; squares represent *Okeania*; triangles represent *Symploca*. Circles represent two or more genera. Molecules without a common name are represented by the ClassyFire most detailed structure type prediction (listed in Appendix A, sheet 2). For the predicted structures of compounds without a common name, see Appendix A.

**Table 1 marinedrugs-19-00020-t001:** Summary statistics and metadata for the 24 high quality genomes obtained in this characterization of tropical filamentous marine cyanobacteria. All genomes have at least 98% CheckM completeness. For more detailed statistics and other metadata, refer to Appendix A, sheet 1. “ID” = identifier; “Frag.” = fragmented; “Comp.” = complete; “BGC” = biosynthetic gene cluster. Most “rare” genomes from Mash analysis are colored in red.

	Collection Code	Genome ID	# of Scaffolds	Frag./Comp. BGC	Taxonomic ID	NCBI Accession
1	ISB3NOV948BCUL	SIOISBB	3	0/11	*Leptolyngbya*	JAAHII01
2	ASI16JUL142CUL	SIOASIH	25	5/41	*Moorena*	JAAHIH01
3	NAC09DEC082ENV	SIO1I7	110	5/12	*Okeania*	JAAHGF01
4	SPB31JAN131ENV	SIO3A2	132	23/8	*Moorena*	JAAHHC01
5	NAC18DEC082ENV	SIO3A5	164	17/11	*Moorena*	JAAHHD01
6	ASI16JUL149ENV	SIO1A7	183	6/8	*Oscillatoria*	JAAHFN01
7	PAB18MAY117ENV	SIO2C2	184	8/6	*Okeania*	JAAHGM01
8	PAB05APR064ENV	SIO1H4	192	11/3	*Okeania*	JAAHGB01
9	PRL23MAR111ENV	SIO1F9	198	5/6	*Okeania*	JAAHFW01
10	PRM25MAR112ENV	SIO1G6	205	20/12	*Moorena*	JAAHFZ01
11	ASY22JUL141ENV	SIO1C2	246	16/11	*Symploca*	JAAHFP01
12	PAB17MAY117ENV	SIO2B3	250	14/2	*Okeania*	JAAHGH01
13	PAB07APR054ENV	SIO1H5	263	13/3	*Okeania*	JAAHGC01
14	PAB03APR065ENV	SIO1H2	279	11/2	*Okeania*	JAAHGA01
15	PAP14JUN083ENV	SIO3C6	290	6/11	*Symploca*	JAAHHJ01
16	PAC17FEB109ENV	SIO2C9	318	14/8	*Okeania*	JAAHGO01
17	PAL11AUG091ENV	SIO2F5	334	18/1	*Okeania*	JAAHGU01
18	PAL01AUG091ENV	SIO2G4	339	16/3	*Okeania*	JAAHGW01
19	PNG21MAY053ENV	SIO2E9	384	16/5	*Symploca*	JAAHGS01
20	PLP20MAR122ENV	SIO3E8	387	23/7	*Moorena*	JAAHHM01
21	PNG22APR061CUL	SIO3F7	399	22/7	*Moorena*	JAAHHP01
22	ASG15JUL146CUL	SIO3F2	454	10/0	*Spirulina*	JAAHHN01
23	PAB17MAY115ENV	SIO2C1	471	25/4	*Symploca*	JAAHGL01
24	ASX22JUL142CUL	SIO1E4_02	12	3/17	*Leptolyngbya*	JAAHFU01

## Data Availability

The data and algorithm generated by this study are available at https://github.com/tiagolbiotech/cyanobiome. The genomes can be found at the following NCBI accession numbers: JAAHII01, JAAHIH01, JAAHGF01, JAAHHC01, JAAHHD01, JAAHFN01, JAAHGM01, JAAHGB01, JAAHFW01, JAAHFZ01, JAAHFP01, JAAHGH01, JAAHGC01, JAAHGA01, JAAHHJ01, JAAHGO01, JAAHGU01, JAAHGW01, JAAHGS01, JAAHHM01, JAAHHP01, JAAHHN01, JAAHGL01, JAAHFU01 (see Appendix A for specific organisms). The *Moorena* genomes are posted as *Moorea* since the name change was made after genome uploads occurred. LCMS files are available at the MassIVE dataset MSV000085210 (file names at Appendix A, sheet 1). The Paired Omics Data Platform identifier 864909ec-e716-4c5a-bfe3-ce3a169b8844.1 contains both genomes and LCMS files (named according to Appendix A, sheet 1).

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
