# Peer review of "A Multi-Omics Characterization of the Natural Product Potential of Tropical Filamentous Marine Cyanobacteria"

_marinedrugs, 2021, doi:10.3390/md19010020_

Round 1

Reviewer 1 Report

This manuscript outlines a multi-omics approach combining genomics and metabolomics analysis of cyanobacteria with the aim of expanding our understanding of natural products from these sources. The authors examine 1923 genomes (most publicly available) including assembling 24 marine cyanobacterial genomes assembled as part of this project. Of these 24 strains, molecular networking analysis was carried out on a subset.

The authors’ conclusions are interesting, demonstrating both that much remains unknown about cyanobacterial metabolites but also that we are potentially reaching a plateau point on diversity in terms of novelty, but that significant diversity still exists within known GCF. The authors demonstrate that using beta diversity in a multi-omics approach is one metric that can help identify these strains of interest. The paper is also an excellent example of the power of some of the newer tools associated with the widely-used GNPS.

This manuscript will be of broad interest to the community.  The data supports the author’s conclusions, the manuscript is well-written, and the figures clear.  I recommend it be published without changes.

Author Response

Reviewer 1 did not request any changes.

Reviewer 2 Report

The authors have presented a combined omics analysis of cyanobacteria using metabolomics and genomics.  Overall, this manuscript highlights several important challenges with regard to genomics of cyanobacteria.  Also, the manuscript provides a nice discussion with regard to the makeup of publicly available genomes.  That combination establishes the current landscape nicely.  When necessary, the authors also employed long-read sequencing to improve the quality of the genomes.  Overall, this is important work and will be of interest to many in the field of natural products.

There are a few things that should be clarified:

  1.  The abstract discusses BiG-SCAPE phylum-wide analysis that revealed the most promising strains for natural product discovery.  While it is clear that those results are conveyed in Figure 2, it isn't exactly clear how the cutoffs were selected.  Is there background information on this?  There are no citations or much discussion this aspect of things.  Perhaps this is clear for people in the genomics field, but as a non-expert this is difficult to follow.  An expanded discussion on this point would be helpful since those cutoffs lead to the PCoA plot.
  2. For Figure S4, it isn't clear where these structures come from.  The paper refers to Dataset S1, sheet 2.  I cannot figure out what that is referring to. I have looked at the paired omics data, but it isn't clear where sheet 2 is or what Dataset S1 is.  As such there is a lot of information around lines 248 to 255 that is unclear.  
  3. Line 249:  "long IUPAC names were replaced by their most detailed ClassyFire prediction"  This is a bit unclear what this means.  Maybe if I could find Dataset S1, sheet 2, it would be more clear. 
  4. For the general reader, it would be helpful to talk a bit about ClassyFire. Along the same lines, the details surrounding data analysis seem a bit sparse with regard to GNPS Molecular Networking.  I think it would be very challenging to reproduce the work in this paper with the limited information in section 4.6 of methods.  A supplemental information portion of this paper would be quite helpful. In particular, a little more detail about how ClassyFire was used and an example of the output? Again a supplemental figure/table that gave a sample output would be helpful.  My sense is that MolNetEnhancer was used which incorporates ClassyFire predictions and not that it was used separately, which is what it sounds like in the manuscript (section 4.6).  

Author Response

Please find our answers to reviewer 2’s concerns below. The text that has been modified and re-submitted (modifications in the text are highlighted in yellow and track changes were on).

Please let us know if you need anything else?

Sincerely,

Tiago Leao and Lena Gerwick

Reviewer 2:

There are a few things that should be clarified:

  1. The abstract discusses BiG-SCAPE phylum-wide analysis that revealed the most promising strains for natural product discovery.  While it is clear that those results are conveyed in Figure 2, it isn't exactly clear how the cutoffs were selected.  Is there background information on this?  There are no citations or much discussion this aspect of things.  Perhaps this is clear for people in the genomics field, but as a non-expert this is difficult to follow. An expanded discussion on this point would be helpful since those cutoffs lead to the PCoA plot.

Answer to 1: Thank you for the comment, we are sorry for not being clear about the cutoff selection. As described in lines 162-169 (original file), we used MIBiG BGCs that contained known structures to perform a cutoff calibration. By inspecting different cutoffs, depicted on the x axis of Figure 2a, we observed the average structural similarities (Figure 2a, y axis) between two or more MIBiG BGCs that were networked together in the same family. We expanded the discussion to make it clear how we selected the cutoff for the gene cluster networking and PCoA (lines 162-171).

  1. For Figure S4, it isn't clear where these structures come from.  The paper refers to Dataset S1, sheet 2.  I cannot figure out what that is referring to. I have looked at the paired omics data, but it isn't clear where sheet 2 is or what Dataset S1 is.  As such there is a lot of information around lines 248 to 255 that is unclear. 

Answer to 2: We apologize if the Dataset S1 got lost in the process. The structures come from enhancing a GNPS classical molecular network with MolNetEnhancer. For such, we had to run separately (for the same set of LC-MS/MS files) the tools for in silico prediction, NAP and Dereplicator+. Once these two in silico analysis are complete, we can run MolNetEnhancer that uses ClassyFire on the predicted structures from both GNPS library hits and the in silico tools to derive the chemical ontology for each structure. MolNetEnhancer will produce an output-file that contains a network very similar to Figure 4, with node colors pre-populated according to the predicted ClassyFire SuperClass that was inferred using the aforementioned structures. The nodes table of the MolNetEnhancer network contains the full ClassyFire predicted ontology, predicted structure and other GNPS information. This node table was pasted in Dataset S1, sheet two. We improved the explanation (lines 246-253) and added the above description to section 4.6 (lines 447-460).

  1. Line 249:  "long IUPAC names were replaced by their most detailed ClassyFire prediction"  This is a bit unclear what this means.  Maybe if I could find Dataset S1, sheet 2, it would be more clear.

Answer to 3: We agree that the use of the Dataset S1 would make this issue more clear. The compounds in the GNPS library that do not contain common name (e.g. veraguamide, apramide, dolastatin, etc) only had long IUPAC names such as “NCGC00385661-01!3,5,7,8-tetramethoxy-2-(3,4,5-trimethoxyphenyl)chromen-4-one”, which we decided to use its ClassyFire most detailed structure prediction such as “chromone” as the common name for the metabolite. We replaced “long IUPAC name” for “molecules without a common name” (lines 263-266 and Figure 4 legend).

  1. For the general reader, it would be helpful to talk a bit about ClassyFire. Along the same lines, the details surrounding data analysis seem a bit sparse with regard to GNPS Molecular Networking.  I think it would be very challenging to reproduce the work in this paper with the limited information in section 4.6 of methods.  A supplemental information portion of this paper would be quite helpful. In particular, a little more detail about how ClassyFire was used and an example of the output? Again, a supplemental figure/table that gave a sample output would be helpful.  My sense is that MolNetEnhancer was used which incorporates ClassyFire predictions and not that it was used separately, which is what it sounds like in the manuscript (section 4.6).

Answer to 4: We agree that the section 4.6 is rather short, we extended the section with more details about the methodology and we included the molecular networking link for the jobs we ran.  A job can be cloned to generate a new job with the same parameters that can be modified by the user or not, reproducing our results. We also included the network file in the github repository (in the folder outputs). We hope that providing access to the jupyter notebooks (in the GitHub repository), cyanobacterial genomes (all deposited in the NCBI database), the LCMS files (in the MassIVE database) and network job (in the GNPS platform), our work here can be easily reproduced.